# Effect of SARS-CoV-2 infection in neonates or in pregnancy on developmental outcomes at 21–24 months (SINEPOST): study protocol for a prospective cohort study

Kathryn Woodward [1], Rosie P Cornish,[2] Chris Gale [3], Samantha Johnson,[4] Marian Knight,[5] Jenny Kurinczuk,[5] Ela Chakkarapani [1]

For numbered affiliations see end of article.

**Correspondence to**
Dr Ela Chakkarapani; Ela. Chakkarapani@bristol.ac.uk

## ABSTRACT

**Introduction** Exposure to SARS-CoV-2 during pregnancy or in the neonatal period may impact fetal or neonatal brain development either through direct central nervous system infection or indirectly through the adverse effects of viral infection-related inflammation in the mother or newborn infant. This study aims to determine whether there are early neurodevelopmental effects of SARS-CoV-2 infection.

**Methods and analysis** We will conduct a prospective national population-based cohort study of children aged 21–24 months who were born at term (≥37 weeks' gestation) between 1 March 2020 and 28 February 2021 and were either antenatally exposed, neonatally exposed or unexposed (comparison cohort) to SARS-CoV-2. Nationally, hospitals will identify and approach parents of children eligible for inclusion in the antenatally and neonatally exposed cohorts using information from the UK Obstetric Surveillance System (UKOSS) and British Paediatric Surveillance Unit (BPSU) national surveillance studies and will identify and approach eligible children for the comparison cohort through routine birth records. Parents will be asked to complete questionnaires to assess their child's development at 21–24 months of age. Outcome measures comprise the Ages and Stages Questionnaire, Third Edition (ASQ-3), Ages and Stages Questionnaire Social-Emotional, Second Edition (ASQ-SE-2), Liverpool respiratory symptoms questionnaire and questionnaire items to elicit information about healthcare usage. With parental consent, study data will be linked to routine health and education records for future follow-up. Regression models will compare ASQ-3 and ASQ-SE-2 scores and proportions, frequency of respiratory symptoms and healthcare usage between the exposed and comparison cohorts, adjusting for potential confounders.

**Ethics and dissemination** Ethics approval was obtained from the London-Westminster Research Ethics Committee. Findings will be disseminated in scientific conference presentations and peer-reviewed publications.

**ISRCTN registration number** ISRCTN99910769.

## WHAT IS ALREADY KNOWN ON THIS TOPIC

⇒ The incidence of vertical transmission of SARS-CoV-2, as well as the incidence of SARS-CoV-2 infection in neonates is low.
⇒ Most neonates infected by SARS-CoV-2 had mild symptoms or were asymptomatic.
⇒ SARS-CoV-2 has neuroinvasive potential and may impact children's neurodevelopment.

## WHAT THIS STUDY ADDS

⇒ A population-based evaluation of the impact of SARS-CoV-2 infection in pregnancy or in the neonatal period on children's neurodevelopment at 21–24 months of age.
⇒ An assessment of healthcare usage and respiratory symptoms in children exposed to SARS-CoV-2 infection as a fetus or neonate compared with non-exposed children.

## HOW THIS STUDY MIGHT AFFECT RESEARCH, PRACTICE AND/OR POLICY

⇒ The outcomes of this study will inform public health advice provided by the National Health Service and clinicians to pregnant women and parents of newborn babies exposed to SARS-CoV-2 infection.

## INTRODUCTION

The SARS-CoV-2 virus has infected over 300 million people worldwide.[1] Changes occurring to the immune system during pregnancy may predispose pregnant women to SARS-CoV-2 infection,[2] and pregnant women with SARS-CoV-2 infection are at increased risk of requiring critical care.[3 4] In a previous study, 95% of babies born to women with confirmed SARS-CoV-2 during pregnancy did not have a confirmed SARS-CoV-2 infection in the neonatal period.[3] Where babies were infected in the neonatal period, possible transmission occurred from the mother in 26%, but this was most commonly in the

postnatal period and vertical transmission appears to be rare.[5] Overall the incidence of SARS-CoV-2 infection in neonates is low (5.6 per 10 000 live births),[5] and most neonates had mild symptoms or were asymptomatic.[6] However, the long-term outcomes for these infants are unknown.

There are various possible mechanisms through which SARS-CoV-2 infection may adversely impact the developing fetus. SARS-CoV-2 infection triggers inflammation[7] which can pass through the placenta and may impact fetal brain development.[8] Severe SARS-CoV-2 infection requiring critical care, which is more common in pregnancy, may impair uteroplacental blood flow inducing ischaemic brain injury to the fetus.[9] SARS-CoV-2 may directly affect the fetal brain, as the placenta expresses the ACE-2 receptor for SARS-CoV-2,[10 11] and SARS-CoV-2 has been detected in placental tissue[11–13] and amniotic fluid.[14] Transplacental transmission has also been accompanied with neurological symptoms.[13 15] However, based on the criteria for probable or possible intrapartum acquired neonatal SARS-CoV-2 infection,[16] only 3% of hospitalised neonates were considered to have possible vertical transmission.[5]

SARS-CoV-2 has been associated with increased incidence of neurological and cognitive deficits in adults.[17 18] In adults with confirmed SARS-CoV-2 infection, SARS-CoV-2 RNA has been detected in brain tissue,[19] and neuroimaging abnormalities have been observed in brain regions functionally connected to the human olfactory system.[20 21] Neuroimaging abnormalities have also been detected in white matter in an infant,[15] and acute disseminated encephalomyelitis-like changes have been observed in the brains of children with acute and subacute SARS-CoV-2 infection.[22] SARS-CoV-2 also evokes a widespread hyperinflammatory state with a 'cytokine storm' in some severely affected individuals,[7] and severe disease is well described in pregnancy.[3 4] In utero or early life exposure to high levels of inflammation[23 24] is known to be associated with cerebral white matter damage, leading to neurodevelopmental disorders in childhood and a range of long-term disabilities.[25] Taken together, this suggests SARS-CoV-2 exposure in a fetus or neonate has the potential to adversely affect long-term neurodevelopment, either directly or indirectly via the effect of inflammation accompanying SARS-CoV-2 infection on the developing brain.

In the UK in March to April 2020, the majority of pregnant women who were hospitalised with SARS-CoV-2 were admitted in the late second or third trimester[3] and 73% gave birth at 37 weeks' gestation or later.[5] Among neonatal cases of SARS-CoV-2 infection, two-thirds occurred in term-born babies.[5] Enhanced developmental surveillance is recommended for preterm but not term-born infants; therefore, there is an urgent need to investigate the potential impact of antenatal or neonatal SARS-CoV-2 exposure on neurodevelopment in term-born infants. Given that SARS-CoV-2 infection impacts the respiratory system during acute infection in both

neonates and adults,[5 26] there may also be a long-term effect on lung function and consequently healthcare usage. Therefore, in addition to assessing neurodevelopment, we intend to establish whether there are any long-term impacts on respiratory function and healthcare usage.

## Study aims and objectives
This study aims to investigate neurodevelopmental effects of SARS-CoV-2 infection.

The primary objective of the study is to investigate the impact of antenatal and neonatal exposure to SARS-CoV-2 on infants' global development at 21–24 months of age.

The secondary objectives are to assess the impact of antenatal and neonatal SARS-CoV-2 exposure on:
► Infants' cognitive, fine motor, gross motor, communication and social-emotional development at 21–24 months of age.
► Infants' respiratory symptoms at 21–24 months of age, and infant's utilisation of healthcare services.

## METHODS
### Study design
A prospective national population-based cohort study of children aged 21–24 months with (exposed cohort) and without (comparison cohort) antenatal or neonatal exposure to SARS-CoV-2, who were born at term between 1 March 2020 and 28 February 2021.

The antenatal exposure cohort will include infants born to women with confirmed SARS-CoV-2 infection during pregnancy and will be identified from the UKOSS COVID-19 in pregnancy study.[3] The neonatal exposure cohort will include infants with confirmed SARS-CoV-2 infection as a neonate and will be identified from the BPSU Neonatal Complications of Coronavirus Disease study.[5] These studies reported ongoing nationwide hospitalisation of pregnant women and neonates with SARS-CoV-2 infection, respectively. The comparison cohort will include infants born to women without evidence of SARS-CoV-2 infection during pregnancy, and whose baby did not have confirmed SARS-CoV-2 infection as a neonate, and these infants will be identified at the same hospitals that reported cases of maternal SARS-CoV-2 infection in pregnancy through the UKOSS study.

### Inclusion criteria
► Antenatal exposure cohort includes term-born (≥37 weeks' gestation) singleton infants of mothers hospitalised with confirmed SARS-CoV-2 infection in pregnancy between 14 and 36+6 weeks' gestation.
► Neonatal exposure cohort includes term-born singleton infants with confirmed SARS-CoV-2 infection who were hospitalised in the first 28 days after birth.
► Comparison cohort includes term-born singleton infants born during the same period as the exposed cohorts, and with no evidence of maternal SARS-CoV-2

infection in pregnancy or neonatal SARS-CoV-2 infection.

## Exclusion criteria
► Children born preterm.
► Children with a major congenital anomaly.
► Children born following multiple pregnancy.
► Children who have left the UK.

## Recruitment
The parents of eligible children will be approached and recruited through the hospitals that cared for the mother or child during SARS-CoV-2 infection or during birth. The study coordinator based at the National Perinatal Epidemiology Unit at the University of Oxford will send study information packs to the hospitals, where one study pack will be dispatched to the parent of each eligible child. The study information packs include:
1. Study invitation letter to parents of the eligible children.
2. Parent information sheet.
3. Form providing information in seven non-English languages (Arabic, Gujarati, Hindi, Polish, Punjabi, Somali and Urdu) on how to find out more about the study in different languages.
4. Informed consent form.
5. Non-participation form.
6. Study entry questionnaire.
7. Freepost return envelope.

The local hospital research teams will be provided with the UKOSS ID for the mothers of antenatal exposure cohort participants and the BPSU ID, National Health Service (NHS) number and date of birth of the neonatal exposure cohort participants. The hospital research teams will use these details to identify the names and addresses of the mothers and eligible children in the antenatal and neonatal exposure cohorts, respectively, from locally held information. The hospital research teams will also identify up to four babies (depending on current recruitment rates) who met the comparison cohort eligibility criteria and were born around the same time as each child with antenatal exposure using routine birth records—two babies born before and two babies born after the antenatal exposure baby, preferably on the same day. The hospital teams will also have the option to telephone the parents of eligible children and inform them about the study prior to posting each parent a study information pack.

## Informed consent
Parents will be able to enrol their children onto the study by completing the consent form and study entry questionnaire and returning these forms to the central research team at the University of Bristol using the freepost return envelope. Parents who do not want to be involved in the study will be able to indicate this by returning the non-participation form. Parents will also be able to complete these forms online by following a weblink provided in the invitation letter. Parents who do not return any forms after 3 weeks will be sent one reminder study pack.

In addition to seeking parental consent for the child to take part in the study, the informed consent form will request consent to (1) retain the contact details of the parents to allow the research team to contact them to be invited to participate in future follow-up assessments, (2) link the child's study data to records held by the Department for Education on academic attainment and special education needs (in the National Pupil Database) and (3) link the child's identifier to NHS Digital records (Hospital Episode Statistics and primary care attendance data).

## Procedures and data collection
Data will be collected through parent-completed questionnaires, either by parents returning paper copies to the central research team by post or by completing the questionnaires online using a secure weblink or over the phone with the study researcher.

Parents will complete the study entry questionnaire when providing informed written consent. This questionnaire will collect information about:
► Parents' preferred choice of completing questionnaires (ie, online or via post or phone).
► The need to use translation services.
► Demographic characteristics.
► SARS-CoV-2 infection status.
► Information on other conditions that might impact neurodevelopment.

Parents will then be sent data collection materials when their child reaches 21–23 months of age. These will consist of the following questionnaires:
1. The Ages and Stages Questionnaire, Third Edition (ASQ-3) 22-month questionnaire[27] in English, a widely used developmental screening tool to identify children at risk of developmental delay.[28] The ASQ-3 assesses development in five domains: fine motor, gross motor, communication, personal-social functioning and problem-solving.
2. The Ages and Stages Questionnaire: Social-Emotional, Second Edition (ASQ-SE-2) 24-month questionnaire[29] in English, a widely used measure to identify children at risk of social-emotional problems.[30]
3. Liverpool Respiratory Symptoms Questionnaire (LRSQ), developed to assess patterns of wheezing and other respiratory symptoms in infants and preschool children.[31]
4. A non-validated questionnaire collecting information about the use of healthcare services and personal financial costs (online supplemental appendix 1).

We chose the ASQ-3 and ASQ-SE-2 as they are validated parent-completed questionnaires that are widely used to identify children at risk of development delay in multiple domains.[28] Questionnaires will be completed over the telephone with the assistance of an interpreter for families who do not speak or read English sufficiently to complete the English version of the questionnaires. Parents who

do not return the questionnaires within 2 weeks will be sent a reminder email or text message, and if there is no response after a month, parents will be contacted again. Parents who do not respond to invitations to complete the ASQ-3 before their child reaches 23 months old will be sent the subsequent age band ASQ-3 24-month or 27-month questionnaire. Parents will be informed of the results of their child's ASQ questionnaire results after completion. If the questionnaires highlight possible developmental delay according to standardised cut-offs, parents will be advised to contact their health visitor and/or general practitioner for further assessment.

### Outcome measures

The primary outcome measure for this study is the mean total ASQ-3 score, and the secondary outcomes are the following:

1. Proportion with one or more ASQ-3 domain scores below the established cut-offs which identify possible developmental delay.
2. Proportion of ASQ-3 domain scores below the established cut-offs which identify possible developmental delay.
3. Mean ASQ-SE-2 total score and proportion of ASQ-SE-2 total scores above the established cut-offs which identify possible developmental delay.
4. Mean LRSQ total score.
5. Frequency of healthcare usage.

### Sample size

The study sample size aims to be able to detect a difference in neurodevelopment associated with perinatal SARS-CoV-2 exposure of similar magnitude to that seen in term-born infants with mild neonatal encephalopathy: a difference in developmental score of 6 points (0.5 SD).[32] Therefore, to detect a similarly clinically meaningful between-group difference of 0.5 SD in mean ASQ-3 total score (mean (SD) 203.02 (50.84))[33] in exposed versus comparison infants, with a type-I error of 0.05 and 90% power, we will need a minimum of 85 infants in each of the two exposure groups and in the comparison group.

To achieve a sufficient sample size for future follow-up studies at school-age considering potential attrition, we aim to recruit at least 200 children in each group. Given the 20% response rate in a previous study of similar design using the UKOSS platform,[34] we will approach the parents of up to 1000 children in the antenatal exposure cohort (currently there are around 830 eligible antenatal exposure cohort children) and up to 2000 families for the comparison cohort, as we expect response rates to be lower for the comparison cohort. Given that there are around 120 eligible children in the neonatal exposure cohort to date, we will approach the parents of all these infants. Consequently, our target sample size comprises 200 children in the antenatal exposure cohort, 200 in the comparison cohort and up to 120 children in the neonatal exposure cohort.

Patient and public involvement

The study design was discussed with a patient and public involvement (PPI) group consisting of a parent of an infant exposed to SARS-CoV-2 in the antenatal period and another parent whose infant was exposed to SARS-CoV-2 in the neonatal period. The PPI group agreed that the research question is important and the design and outcomes to be measured are appropriate. The parent information sheet and consent form were developed with input from the PPI group, and further PPI will be sought to support the conduct of the study and help with interpreting the results.

### Data analysis

We will use descriptive characteristics to compare the exposed and comparison cohorts and assess their representativeness (against national surveillance data, and data from the Office for National Statistics and National Records for Scotland, for the exposed and comparison cohorts, respectively). We will use multiple linear and logistic regression to compare primary and secondary outcomes between the exposed and comparison cohorts, adjusting for potential confounders (eg, baseline sociodemographic factors including ethnicity, sex, socioeconomic status) and carry out exploratory subgroup analyses by gestation for antenatal exposure and severity of SARS-CoV-2 infection in the neonatal period. We will also conduct sensitivity analysis to evaluate the impact of children with other conditions that could impact neurodevelopment including hypoxic-ischaemic encephalopathy, seizures, perinatal stroke, bacterial or non-SARS-CoV-2 viral meningitis and neuroimaging abnormalities in the neonatal period.

## DISCUSSION

The neuroinvasive potential of SARS-CoV-2 was speculated on formulating the study protocol, as some adults with moderate/severe SARS-CoV-2 infection presented with neurological manifestations,[35] raised neuronal injury and glial activation markers,[36] and as the ACE-2 receptor for SARS-CoV-2 is expressed on neurons, glial cells and oligodendrocytes.[37–39] This speculation has since been supported by evidence linking SARS-CoV-2 infection to brain abnormalities and cognitive deficits in adults.[17 18 20 21]

Currently, there is a lack of representative studies examining the impact of maternal SARS-CoV-2 infection in pregnancy and neonatal SARS-CoV-2 infection on the long-term neurodevelopment outcomes of infants. Identifying whether children in these cohorts may be at increased risk of developmental delay is important, as this will inform public health advice provided by the NHS and clinicians to pregnant women and parents of newborn children exposed to SARS-CoV-2 infection.

### Ethics and dissemination

Approval was obtained from London-Westminster Research Ethics Committee (REC; protocol no: 2021-93),

and subsequent amendments to the protocol requiring REC approval were implemented after revised documents were reviewed and approved by the REC and participating organisations. Non-substantial amendments including the addition of new sites were reviewed by the sponsor, submitted to the Health Research Authority (HRA) for information only, and shared with the participating organisations. Findings will be disseminated in scientific conference presentations and peer-reviewed publications.

**Author affiliations**
[1]Translational Health Sciences, Bristol Medical School, University of Bristol, Bristol, UK
[2]Population Health Sciences, Bristol Medical School, University of Bristol, Bristol, UK
[3]Neonatal Medicine, School of Public Health, Faculty of Medicine, Imperial College London, London, UK
[4]Department of Health Sciences, University of Leicester, Leicester, UK
[5]NHIR Policy Research Unit in Maternal and Neonatal Health and Care, National Perinatal Epidemiology Unit, Nuffield Department of Population Health, University of Oxford, Oxford, UK

**Acknowledgements** We also acknowledge the contributions from the co-collaborators on this study: K M Abel, H Mactier, S Doherty, S Ladhani, D Sharkey and E S Draper.

**Contributors** EC, CG, MK, JK, SJ and RPC contributed to the conception and design of the study and KW drafted the manuscript, which was revised critically for important intellectual content by the other contributing authors. All authors approved the final version to be published.

**Funding** The research activities are funded by a project grant from Action Medical Research (GM2905). The BPSU study was funded by the National Institute for Health Research (NIHR) Policy Research Programme, conducted through the NIHR Policy Research Unit in Maternal and Neonatal Health and Care, PR-PRU-1217-21202. The UKOSS study was funded by the NIHR Health Services and Delivery Research Programme (project no 11/46/12). MK is an NIHR senior investigator.

**Patient and public involvement** Patients and/or the public were involved in the design, conduct, reporting or dissemination plans of this research. Refer to the Methods section for further details.

**Patient consent for publication** Not applicable.

**Ethics approval** This study involves human participants and was approved by London-Westminster Research Ethics Committee (protocol no: 2021-93). Participants gave informed consent to participate in the study before taking part.

**Data availability statement** No data are available.

**ORCID iDs**
Kathryn Woodward http://orcid.org/0000-0001-9630-3433
Chris Gale http://orcid.org/0000-0003-0707-876X
Ela Chakkarapani http://orcid.org/0000-0003-3380-047X

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
