## [Reviewer comments · BMJ Paediatrics Open]

ARTICLE DETAILS

TITLE (PROVISIONAL)	The effect of SARS-CoV-2 infection in neonates or in pregnancy on developmental outcomes at 21-24 months (SINEPOST): Study protocol for a prospective cohort study.
AUTHORS	Woodward, Kathryn Cornish, Rosie P Gale, Chris Johnson, Samantha Knight, Marian Kurinczuk, Jenny Chakkarapani, Ela

VERSION 1 – REVIEW

REVIEWER	Reviewer name: Dr. Conrad Kabali Institution and Country: 2264 Spence Lane, Canada Competing interests: None
REVIEW RETURNED	18-Jun-2022

GENERAL COMMENTS	The manuscript is very well written. I recommend it to be published as is.
--

VERSION 1 – AUTHOR RESPONSE

To Professor Imti Choonara, MBChB, MD, FRCPCH, DTM&H
Editor-in-Chief
BMJ Paediatrics Open journal

25th July 2022

Dear Professor Choonara,

Thank you for the opportunity to submit a revised version of our manuscript titled " The effect of SARS-CoV-2 infection in neonates or in pregnancy on developmental outcomes at 21-24 months (SINEPOST): Study protocol for a prospective cohort study." to BMJ Paediatrics Open.

We thank the editors and reviewer for their time and constructive comments on our manuscript. In the following, we address their concerns and feedback point by point.

Editor in Chief comments to Author:

Please add the non-validated questionnaire (for collecting information about the use of health care services and personal financial costs) as an appendix

Reply: Thank you. We have added the non-validated questionnaire as an appendix.

Associate Editors comments to Author:

This is a nice protocol and will contribute valuable new information. I ask for a few minor clarifications:

1. I think it would be helpful to clarify that the ASQ is a developmental screening tool (risk for having developmental delay) not a diagnostic tool. The language in the protocol isn't always precise on this point ("widely used measure to assess children's development"). A sentence or two about why the ASQ was chosen would also be helpful (makes sense since it is a parent reported instrument but the obvious weakness is not a diagnostic instrument).

Reply: Thank you. We have now clarified that the ASQ is a development screening tool in line 10, page 6: "The Ages and Stages Questionnaire®, Third Edition (ASQ-3) 22-month Questionnaire (27) in English; a widely used developmental screening tool to identify children at risk of developmental delay (28)." We have also added a sentence explaining why the ASQ was chosen in line 24, page 6: "We chose the ASQ-3 and ASQ-SE-2 as they are validated parent-completed questionnaires that are widely used to identify children at risk of development delay in multiple domains (28)."

2. For the sample size calculation be explicit about where the mean population ASQ score is from (what population). It might be worth noting that ASQ scores are often affected by the language of administration and so note what languages ASQ will be provided in

Reply: Thank you for your comment. Our primary outcome is the mean total ASQ-3 score (line 39, page 6). As already detailed in the data analysis section, line 23, page 7, we specified that we aim to "compare primary and secondary outcomes between the exposed and comparison cohorts". Our sample size calculation was based on the "mean ASQ-3 total score (mean (SD) 203.02 (50.84))" (line 60, page 6) identified by Steenis et al., 2015 in a cohort of 60 children for which the reference is provided. These data are from a study of 1244 Dutch children whose parents completed the Dutch versions of the ASQ-3 and Bayley-III assessments, to assess the accuracy of using the ASQ-3 as a screening measure for infants and toddlers at risk of a developmental delay.

We have also added that we will be using the English versions of the ASQ-3 and ASQ-SE. Line 10: "The Ages and Stages Questionnaire®, Third Edition (ASQ-3) 22-month Questionnaire (27) in English"; Line 14 "The Ages and Stages Questionnaire®: Social-Emotional, Second Edition (ASQ-SE-2) 24-month Questionnaire (29) in English". Furthermore, we have added into the protocol, line 25, page 6, that "Questionnaires will be completed over the telephone with the assistance of an interpreter for families who do not speak or read English sufficiently to complete the English version of the questionnaires."

3. Sampling procedures for the neonatal exposure cohort are explicit but it is not clear how the databases will be sampled for the other two cohorts (and what rough proportion of those databases 1000 and 2000 represent).

Reply: Thank you. Based upon our most recent data, we have now added into the protocol that "currently there are around 830 eligible antenatal exposure cohort children" (line 10, page 7). Therefore, to date we aim to approach up to 830 children in the antenatal exposure cohort. The total number of antenatal exposure cohort participants who will be approached depends on the number of NHS trusts who are able to take part in the SINEPOST study, and the number of eligible antenatal exposure cohort participants at each of these trusts. However, our maximum number of antenatal exposure cohort participants to be approached was set to 1000.

For the comparison cohort, "infants will be identified at the same hospitals that reported cases of maternal SARS-CoV-2 infection in pregnancy through the UKOSS study." (line 33, page 4), and the hospital research teams will "identify up to four babies who met the comparison cohort eligibility criteria and were born around the same time as each child with antenatal exposure" (line 23, page 5). The total number of comparison cohort participants identified and approached by each NHS trust will depend on current recruitment rates, which is expected to be lower than the exposure cohort recruitment rates. This has now been included on line 23 page 5 which states "The hospital research teams will also identify up to four babies (depending on current recruitment rates) who met the comparison cohort eligibility criteria" for each child with antenatal exposure. We have also amended line 11 page 7 to say we will approach "up to 2000 families for the comparison cohort, as we expect response rates to be lower for the comparison cohort." Therefore, the maximum number of comparison cohort participants to be approached was set to double the number of antenatal exposure cohort participants to be approached.

Reviewer's comments:

The manuscript is very well written. I recommend it to be published as is.

Reply: Thank you for your comment.

We look forward to hearing from you in due course.

Yours sincerely,

Dr Ela Chakkarapani, MD Dr Kathryn Woodward, PhD
Consultant Senior Lecturer in Neonatology Research Associate in Translational Health Sciences